# Novel Design of an α-Amylase with an N-Terminal CBM20 in *Aspergillus niger* Improves Binding and Processing of a Broad Range of Starches

**DOI:** 10.3390/molecules28135033

**Published:** 2023-06-27

**Authors:** Andika Sidar, Gerben P. Voshol, Erik Vijgenboom, Peter J. Punt

**Affiliations:** 1Institute of Biology Leiden, Leiden University, 2333 BE Leiden, The Netherlands; g.voshol@genomescan.nl (G.P.V.); vijgenbo@biology.leidenuniv.nl (E.V.); 2Department of Food and Agricultural Product Technology, Gadjah Mada University, Yogyakarta 55281, Indonesia; 3GenomeScan, 2333 BZ Leiden, The Netherlands; 4Ginkgo Bioworks, 3704 HE Zeist, The Netherlands

**Keywords:** α-amylase, glycoside hydrolase family 13, carbohydrate-binding module, *Aspergillus niger*, starch binding purification, raw starch hydrolysis

## Abstract

In the starch processing industry including the food and pharmaceutical industries, α-amylase is an important enzyme that hydrolyses the α-1,4 glycosidic bonds in starch, producing shorter maltooligosaccharides. In plants, starch molecules are organised in granules that are very compact and rigid. The level of starch granule rigidity affects resistance towards enzymatic hydrolysis, resulting in inefficient starch degradation by industrially available α-amylases. In an approach to enhance starch hydrolysis, the domain architecture of a Glycoside Hydrolase (GH) family 13 α-amylase from *Aspergillus niger* was engineered. In all fungal GH13 α-amylases that carry a carbohydrate binding domain (CBM), these modules are of the CBM20 family and are located at the C-terminus of the α-amylase domain. To explore the role of the domain order, a new GH13 gene encoding an N-terminal CBM20 domain was designed and found to be fully functional. The starch binding capacity and enzymatic activity of N-terminal CBM20 α-amylase was found to be superior to that of native GH13 without CBM20. Based on the kinetic parameters, the engineered N-terminal CBM20 variant displayed surpassing activity rates compared to the C-terminal CBM20 version for the degradation on a wide range of starches, including the more resistant raw potato starch for which it exhibits a two-fold higher Vmax underscoring the potential of domain engineering for these carbohydrate active enzymes.

## 1. Introduction

α-Amylases are enzymes that catalyse the endo-amylolytic cleavage at 1,4-glucosidic bonds of starch and find application in a broad range of industrial processes using starch as substrate [1,2]. The α-amylase secreted by *A. niger* is a member of the glycoside hydrolase family 13 (GH13) as classified by the Carbohydrate-active enzymes (CAZymes) database [see CAZy database at www.cazy.org (accessed on 25 February 2023)] [3]. Many of the *A. niger* GH13 amylases have an ancillary starch binding domain (SBD) called carbohydrate-binding module family 20 (CBM20), always attached at the C-terminus [4]. 

Currently there are 15 CBM families listed in the CAZy database that have a starch binding activity and are considered to support enzymatic activity by increasing proper attachment on the polysaccharide [5]. Among them, CBM20 is the best studied and first discovered SBD, identified at the C-terminal domain of *Aspergillus niger* glucoamylase [6,7]. The CBM20 domains bring special interest as they have been reported to play a role not only in binding but also in the disruption of the surface of starch granules, making the substrate more accessible to be cleaved by starch degrading enzymes and perform degradation at a higher rate [8,9]. This is reported for *A. niger* glucoamylase [10,11], *Cryptococcus* sp α-amylase [12], and AA13 polysaccharide monooxygenases of *Neurospora crassa* [13].

Starch is a glucose polymer that occurs in nature in semi-crystalline granules consisting of linear glucose polymers with 1–4 linkages (amylose) and branched polymers containing both 1–4 and 1–6 bonds (amylopectin). Variation in the amylose:amylopectin ratio plays an important role in controlling the size and shape of the starch granules. A higher amylose content and a denser structure of the granule lead to an increased resistant starch fraction [14,15]. The amylose:amylopectin ratio and granule size are variable, depending on the plant species as well as the region where a crop was grown since environmental or climate conditions could affect the type of starch granule [16,17]. For example, in potato starch the average granule size ranges from 20 to 110 μm, in maize starch the granules range from 15 to 20 μm, and in rice starch the granules on average range from 3 to 5 μm in size [18,19]. These variations in granule size and polymer composition affect the susceptibility of the starch towards decomposition by starch degrading enzymes. Furthermore, the carbohydrate polymers in the starch granules are present in a complex with small amounts of non-carbohydrate components, such as lipids, proteins, and phosphate, that also affect starch processing by enzymatic hydrolysis with enzymes such as α-amylase. Granule complexity is considered to be the main barrier for enzymes to access all types of starches [20]. Therefore, the improvement of substrate binding and hydrolysis is an important goal in reaching the full decomposition of starch-based substrates.

Enzyme engineering has led to the discovery of improved enzymes for a more efficient hydrolysis of substrates, in particular recalcitrant polysaccharides. Several enzyme modification strategies through molecular engineering have been applied to create improved enzymes, including mutagenesis, linker modification, and truncation, as well as N or C-terminal fusions [21,22,23,24]. In particular the fusion of CBM to the catalytic domain of starch degrading enzymes has been shown to be a promising approach to improve enzyme activity towards recalcitrant substrates. In general, in protein domain architecture, the CBM domains can be found either at the N- or C-terminus of the catalytic domain of an enzyme, and for some CAZyme classes both configurations are known. In the case of CBM20, the C-terminal position is much more prevalent than the N-terminal position in GH13 α-amylase and usually exists as a single copy [4,5]. CBM20 is present in tandem with other CBMs, such as CBM48 and CBM34, at the N-terminus of a GH13 from *Bacillus* sp. AAH-31 [25] and reviewed in [5]. This enzyme was characterised as α-amylase based on its ability to degrade soluble starch [26]. However, it was reported later that this α-amylase did not bind to granular starch and that the protein structure of the catalytic domain is more similar to that of neopullulanase [27,28]. Although we focus on characterised CBM20 containing α-amylases, it should be noted that few GH13 amylases have an N-terminal CBM21, a CBM closely related to the CBM20 family [29,30].

In xylanase engineering experiments it has been shown that the fusion of the C-terminal CBM9 from *T. maritima* xylanase to either the C- or N-terminus of *A. niger* GH11 xylanase resulted in an increase in thermostability as well as activity [31]. CBM20 fusions were also found to increase catalytic performance. However, CBM20 tend to be fused always at the C-terminus of amylase, e.g., the catalytic activity of barley α-amylase on starch granules was enhanced by the C-terminal fusion of the CBM20 from *A. niger* glucoamylase [32]. Moreover, replacement of the C-terminal CBM69 in α-amylase (AmyP) with C-terminal CBM20 from *Cryptococcus* sp increased the catalytic efficiency toward raw rice starch [12]. For AA13 LPMOs, the fusion of the CBM20 originally located at the C-terminus of AA13 *Neurospora crassa* on to the C-terminus of AA13 LPMOs from *Myceliophthora thermophila* resulted in a more than 50% increase in amylose binding [33]. However, to date, no research has reported the starch hydrolysis of α-amylases engineered with an N-terminal CBM20.

In *A. niger*, the simplest α-amylase domain organisation consists of a GH13 catalytic domain followed by a Domain of Unknown Function (DUF1966). Some of the *A. niger* amylases also have a CBM20 domain at the C-terminus, however an N-terminal CBM20 is not present in any α-amylases from this organism. Furthermore, according to the current database of protein families and domain organisation available in InterPro [34], approximately 2400 GH13 α-amylases with a C-terminal CBM20 were identified, while only around 30 proteins of the GH13 family carrying a CBM20 at the N-terminus were found, predominantly in algae. To date, none of the GH13 family members that carry CBM20 at the N-terminal position have been functionally characterised. Inspired by the occurrence of the natural domain architecture present in algae, our research focused on exploring the functionality of a new characterised domain organisation of a chimeric α-amylase from *A. niger*, by assembling the GH13 catalytic domain with an N-terminal CBM20 domain. The opportunity that this approach could offer was supported by recent findings that the N-terminal fusion of CBM20 onto 4-α-glucanotransferases from the GH77 family was able to enhance the enzyme’s affinity toward granular starch [35].

Obviously in engineering the enzyme domain architecture, not only the order of domains but also the linker peptides connecting various domains can influence enzyme activity and substrate binding, as is demonstrated in studies showing that variation in linkers affects the kinetic behavior of modular α-amylase [23] and cellulases [36,37], as well as in binding and activity from lytic polysaccharide monooxygenases (LPMO) [38]. Therefore, we considered that the selection of the linker connecting the domains could be essential for generating an active chimeric enzyme with desired properties. The goal of this research was to generate and investigate the unique domain architecture of an *A. niger* α-amylase carrying an N-terminal CBM20. The effects of this new domain architecture on substrate binding properties and catalytic activity as well as kinetic parameters are reported. This research suggests that the domain engineering of α-amylase is a valuable approach to obtain chimeric amylase with improved binding as well as hydrolysis especially with raw starch granules as substrate.

## 2. Results and Discussion

### 2.1. Domain Architecture of α-Amylase Design

In designing a novel configuration of a GH13 α-amylase-containing modular enzyme, not only should the secretion signal as well as various fungal domains such as the activity domain and the substrate binding domain be considered but also the linkers connecting these different domains. In the research presented in this paper, a completely new design of α-amylase was explored. To avoid unwanted effects of de novo designed linkers, we took special care in using native linker configurations. This meant that we had to select configurations derived from different annotated genome sources where these enzyme configurations existed, since no N-terminal CBM20_GH13 amylase has been described. The linker connecting GH13 with CBM20 at the N-terminal position was derived from an uncharacterised algal GH13 family protein with an N-terminal CBM20 (JAC81539.1). However, as the annotated protein encoded by this gene was devoid of a signal sequence, the linker used in joining a signal sequence with the N-terminal CBM20 was derived from the hinge connecting a signal sequence from a fungal laccase from *Baudoinia panamericana*, which also contained a CBM20 at the N-terminus of the catalytic domain (XP_007679364.1). Furthermore, the actual signal sequence as well as the CBM20 and GH13 domain of our chimeric design were derived from *A. niger*.

The signal sequence was retrieved from *A. niger* glucoamylase (glaA), as this was shown to be a versatile secretion signal [39]. Additionally, the CBM20 domain was also retrieved from *A. niger* glucoamylase, representing a well and extensively studied enzyme containing a CBM20 [40,41,42,43,44,45,46,47]. The GH13 catalytic domain was derived from the well-known GH13 α-amylase from *A. niger* CBS 513.88, of which the accurately annotated genome sequence was available [48]. Moreover, the GH13 α-amylase from *A. niger* CBS 513.88 has been widely studied [49,50,51]. This GH13 gene was also used in the design of the GH13 α-amylase without CBM20 (Figure 1). For the design of GH13 with a C-terminal CBM20, the α-amylase gene from *A. niger* AB4.1 was used. In this case we could not use *A. niger* CBS 513.88 as a source as this strain does not have a gene encoding an α-amylase with a C-terminal CBM20 [3,50]. In this research, this GH13_CBM20 was used as a reference α-amylase, and therefore the protein sequence used was kept identical as present in *A. niger* AB4.1. The domain architectures of these three α-amylase variants are shown in Figure 1. The amino acid sequences and the protein alignments are presented in Appendix A, respectively. The protein sequence of GH13 without CBM20 is identical to the GH13 in the engineered domain architecture and is 99% identical to that of native GH13_CBM20, with a full conservation of the catalytic residues Asp, Glu, and Asp [51,52,53] (Appendix A). Similarly, the CBM20 sequences of the engineered and native reference are not completely identical, but the functional starch binding sites are conserved, including the two critical Trp residues as well as two Tyr residues, which are essential in CBM–substrate binding (Appendix A) [13,42].

In modular CAZymes with CBMs, the modules are expected to act cooperatively during catalysis, and that degree of linker flexibility is essential for the movement of the domains. It was reported that the amino acid composition of the linker plays a role in determining the linker flexibility which further influences the interaction between domains [35,53]. Several amino acids, such as Serine (Ser), Threonine (Thr), and Glycine (Gly), are known as small polar amino acids that contribute to providing flexibility, as these residues are usually present in natural linkers [53]. Based on the domain and sequence analysis in the HMMER program (version 3.3.2) against the Pfam database, the linkers connecting modules in native α-amylases with C-terminal CBM20 attachment contain several clustered Ser and Thr residues [54] (Table 1). Besides the role of Ser and Thr in flexibility, these residues could also be involved in O-glycosylation in particular when found in clusters. As shown in Table 1, the engineered N-terminal CBM20_GH13 is devoid of any clustered Ser or Thr residues. Basically, glycosylation does occur in *A. niger*, but we have not determined if the α-amylases analysed in our research were glycosylated. However, this has no effect on the major conclusion of our research, i.e., that an N-terminal CBM20 domain results in a full active amylase.

In the chimeric CBM20_GH13, the natural linker used for connecting the two domains contains a large number of proline (Pro) residues, while the Pro residues are absent in the linker of the native GH13_CBM20 (Table 1). Pro is a unique non-polar amino-acid residue with a cyclic side chain which could play a role in the rigidity as well as the elasticity of the linker, as it was demonstrated that Pro-rich sequences have a rigid and spring-like elastic structure [54,55]. The dynamic feature of Pro residues may be helpful to maintain the stretched elastic conformation, enabling the linker to expand and retract [56,57,58]. Therefore, the presence of Pro residues could be beneficial in either spanning a longer distance or narrowing the space between a domain and its substrates. Moreover, Pro and Ala (and Ser) residues in the linker motif may not only contribute to its flexibility and expansion capacity but may also add to avoiding protein aggregation as well as providing stability against proteolysis [59,60,61]. In addition, the longer size of the linker suggests that it may span a broad range of distances in reaching out for the substrate (Table 1).

### 2.2. Expression of α-Amylase from A. niger Transformants

For selecting the best performing transformants, 25 transformants from each construct were screened for activity on AZCL-amylose plates (see Material & Methods Section 3.5.1). Submerged fermentation was then carried out, with the transformants showing the highest α-amylase activity. Spent medium was collected for further enzyme activity tests using an AZCL-amylose suspension. Both GH13 α-amylase with a CBM20 showed more activity than GH13 alone, indicated by the more intense blue colour released and OD measured from AZCL-amylose degradation (Figure 2). Based on these results, experiments were performed on purification of the amylase proteins for a more detailed comparison of enzyme performance.

### 2.3. Purification of GH13 α-Amylase Variants Based on Starch Binding

Initial purification of the various α-amylases based on starch binding was carried out based on the method described by [49] with modifications to evaluate both the enzymatic activity of the various α-amylase proteins as well as starch binding properties in more detail. As described in the Materials and Method Section 3.6, for this protein purification we used corn starch as it was reported that this type of starch has a spongy surface with numerous small pores and contains lower levels of protein and lipids in the granules, which could otherwise potentially increase the unspecific absorption of α-amylase [15,62]. In this experiment, binding was conducted at 4 °C, where no significant enzymatic activity was detected. During binding, incubation with corn starch was continued until no further reduction in activity was detected in unbound fraction. The evaluation of α-amylase purification was carried out by measuring the activity in the spent medium, the unbound fraction, and the eluted protein fraction using 2-chloro-4-nitrophenyl α-D-maltotrioside (CNPG3) and various starches as substrate (Appendix A). The unbound fraction refers to protein that was not bound to corn starch, while the eluted protein fraction represents the purified enzyme fraction that was collected from the elution of bound protein on corn starch using malto-dextrin-containing buffer. In this measurement, α-amylase activities were detected based on both the release of chloro-nitrophenol from CNPG3 and the measurement of reducing sugar from the starch substrates, as described in the Materials and Methods Section 3.5.2 and Section 3.5.3, respectively. Furthermore, the total activity of α-amylase measured in the spent medium and unbound fraction was used as the basis to evaluate its binding potential with the spent medium activity set at 100% (Figure 3, and Appendix A). The binding capacity value obtained for the various substrates showed a quite consistent value, of which CBM20_GH13 has the highest binding potential with in average 57% bound to corn starch compared to approximately 43% for the GH13_CBM20 configuration (Appendix A). As expected, the GH13 without CBM has a very low binding capacity, of only about 10%. The fact that not all of the enzyme was retained by starch binding may be explained by the fact that during binding the electrostatic interaction of the enzyme and the starch adsorbent influences the adsorption process as reported for the amylase purification from *Bacillus* [63,64].

The enzyme activities recovered in the binding experiment using the parental strain, *A. niger* MGG029-Δ*aamA* (Figure 3), are probably due to the presence of small oligosaccharides such as maltotriose or maltose present in the starch substrates which could be hydrolysed by alpha glucosidases/maltases present in the background. Likewise, the CNPG3 substrate composed of maltotriose and chloro-nitrophenol units could show residual hydrolysis due to alpha glucosidases. However, the activity detected in the background of *A. niger* MGG029-Δ*aamA* toward both starch and CNPG3 substrates was much lower than the activity of the α-amylase produced by transformants carrying any of the three α-amylase genes (Appendix A). Moreover, almost all total activity as detected for the parental strain remained unbound to the corn starch (Figure 3 & Appendix A).

For further enzyme characterisation, elution with soluble malto-dextrin was used to release the enzyme activity bound to the starch. Dextrin is a low molecular weight starch derivative, suitable for competitive binding and thus replacing the more complex starch polymer. The simpler molecule of malto-dextrin is more accessible for the binding of the enzyme compared to the molecule with greater molecular size [65]. In subsequent enzyme characterisation the malto-dextrin was removed from the eluted protein sample as described in the Materials & Methods Section 3.4.

Subsequently, the purified enzyme fractions were analysed with SDS-PAGE and Zymogram analysis. The SDS-PAGE of purified protein revealed essentially a polypeptide band for each purified fraction with the expected molecular mass around 80 kDa for the CBM20_GH13 and GH13_CBM20 and around 55 kDa for GH13. The fact that only a weak band for the GH13 was observed reflects the low binding to corn starch (Appendix A).

Using amylase zymogram analysis to detect the amylases, all purified fractions (EP) showed a single protein band with amylase activity (Figure 4). All strains including the parental host strain used for the expression of the amylases show a slower migrating background activity towards the AZCL-Amylose in the spent medium and unbound fraction, which is absent from the purified fractions, demonstrating that a good level of purity was achieved. As the α-amylase variants have similar predicted isoelectric points, 4.3, 4.2, and 4.1 for CBM20_GH13, GH13, and GH13_CBM20, respectively, we expect these to migrate at roughly the same position in the gel. Taken together, the result shows that both amylases with either a C-terminal or an N-terminal CBM20 can be purified with the starch binding protocol. The chimeric design of amylase with the N-terminal CBM20 displays a similar performance on corn starch as the native α-amylases with a C-terminal CBM.

### 2.4. Enzymatic Activity and Kinetic Parameter of Purified α-Amylases on Various Substrates

Further analysis of the enzymatic activity was conducted for the two CBM20 containing α-amylases obtained after the starch-binding purification, CBM20_GH13 and GH13_CBM20. The protein concentration of both samples was normalised to 0.5 µg, and similar band intensities were confirmed with SDS-PAGE (Appendix A). As shown in Figure 5, the enzymatic hydrolysis of the newly designed CBM20_GH13 toward AZCL-amylose was faster than that of GH13_CBM20. CBM20_GH13 exhibited a specific activity in mg per hour which was higher than that of GH13_CBM20, being 855 U/mg,h and 583 U/mg,h, respectively.

The enzymatic activities of both α-amylases with N- and C-terminal CBM20 were also examined using various substrates, such as CNPG3 and several starches from potato, rice, corn, wheat, and soluble starch. Both enzymes were shown to hydrolyse a broad range of starches and CNPG3 substrates (Appendix A). Similar as found for the binding potential, the N-terminal position of the CBM20 contributes positively to enzymatic activity. The CBM20_GH13 outperforms GH13_CBM20 on all substrates. The largest difference in specific activity was seen on raw potato starch, where CBM20_GH13 performed twice as well as GH13-CBM20 (Table 2).

Furthermore, to allow accurate comparison, the kinetic parameters of CBM20_GH13 and GH13_CBM20 were determined through measuring enzymatic specific reaction rates with various substrate concentrations and calculated in Unit per 1 mg of protein (Appendix A). The values of Michaelis constant (Km) and the maximum reaction rate (Vmax) were measured as mg/mL and µmol/min, respectively. The kinetic profile of starches and CNPG3 hydrolysis were monitored by measuring the released reducing sugar or released chloro-nitrophenyl respectively during hydrolysis by α-amylase.

In all cases, the Vmax values of chimeric amylase with N-terminal CBM20 was higher than the Vmax of GH13_CBM20, especially for the CNPG3 substrate. As expected, the hydrolysis rate of the “simple” artificial chromogenic CNPG3 substrate was more rapid than the other “complex” starch substrates with a Vmax of 90.9 and 71.4 μmol/min for CBM20_GH13 and GH13_CBM20, respectively. CNPG3 is regarded as an easier substrate than the natural starch polymers, which have a much more complex structure, making them less susceptible to degradation by amylase.

Among starches, both α-amylases with CBM20 acting on soluble starch and rice starch showed a higher Vmax compared to the Vmax for other types of starches. Moreover, especially for raw potato starch a very low Vmax was observed. It was reported that rice starch has smaller granule size distribution and is more porous compared to other common cereal starches such as maize and wheat, of which this small particle size and higher porosity provides more available surface for enzyme and water adsorption, affecting the high rate of hydrolysis by α-amylase [66]. Moreover, raw starches such as the potato starch we have used generally have a high amylose content and a low digestibility [67] making them more resistant to α-amylase. However, although a lower Vmax was observed with raw potato starch as substrate, the CBM20-GH13 performed twice as well as GH13-CBM20, 3.2 and 1.5 µmol/min, respectively (Table 3). For all other starches the difference between the Vmax was smaller, but CBM20_GH13 was better in all cases (Table 3).

Furthermore, in terms of enzyme affinity represented by the Km values, in general the CBM20_GH13 showed a lower Km than GH13_CBM20 (Table 3, Appendix A). A low Km value indicates a high affinity in the enzyme for the substrate. Apparently, the combination of linker residue composition and domain architecture influences this behaviour, which is similar to what was reported earlier [36]. As mentioned earlier, the linker connecting GH13 with C-terminal CBM20 in the native α-amylase is predicted to have a more rigid structure. Our results show that the new arrangement of the GH13 α-amylase domain with N-terminal CBM20 was fully active and exhibited efficient starch binding and higher amylase activity compared to the GH13_CBM20.

To explore whether the prediction of the configuration of the modular α-amylase designs used in this research could help to explain the obtained results, 3D modelling of the different proteins was carried out. 3D protein models were built based on the amino acid sequence using Alphafold2. Basically, the GH13 catalytic domain of *A. niger* used in this study exhibited an open structure in the substrate binding cleft (Figure 6), corresponding with the structure of *A. niger* GH13 α-amylase that was reported previously [51,52,53]. Moreover, the Alphafold2 prediction shows that the orientation of the CBM20 domain toward the catalytic module for CBM20_GH13 and GH13_CBM20 is clearly different (Figure 6). In the chimeric CBM20_GH13, the linker connecting the CBM20 appeared from the centre-back side of the GH13 catalytic domain where the N-terminus of GH13 structure is located (Figure 6). Moreover, the CBM20 structure was oriented in a parallel position towards the catalytic domain as well as in a close proximity to the substrate binding cleft of the catalytic module, potentially allowing a synergistic interaction between the modules for accessing and processing substrate (Figure 6). This predicted structure feature suggested an optimal free mobility for the catalytic module in any direction to access the substrate based on the linker length and its position dangling from the center back-side of the GH13 catalytic domain. Meanwhile, in the Alphafold2 prediction of the GH13_CBM20 structure, the linker appeared from the bottom of the DUF1966 domain and tends to place the CBM away from the substrate binding cleft of the catalytic module, locating the CBM20 at the bottom of GH13 domain, suggesting a less efficient interaction between the CBM and catalytic domain to work cooperatively in processing the substrate.

In conclusion, a new gene design of the domain architecture of an *A. niger* GH13 α-amylase with the CBM20 domain at the N-terminus was successfully created and expressed in *A. niger*. Compared with the α-amylase without CBM20, the enzymes with either an N-terminus or C-terminus CBM20 showed a better performance in binding and degradation of a wide range of starches. Furthermore, the amylases carrying a CBM20 could be purified through a starch binding protocol. More detailed kinetic enzyme characterisation revealed that the chimeric CBM20_GH13 showed better Vmax and Km values on various starch substrates compared to GH13_CBM20. Therefore, α-amylase domain engineering can be a promising approach in designing new enzyme configurations with improved hydrolysis.

## 3. Material and Methods

### 3.1. Microbial Strains, Plasmid, Medium, and Substrates

*A. niger* strain MGG029 Δ*aamA pyrE*^−^ was used as the transformation host, while *Escherichia coli* DH5α was used for plasmid propagation. *A. niger* AB4.1 was used as a source for the genomic DNA used as a DNA template for obtaining the native GH13 α-amylase with the C-terminal CBM20. Plasmid pMA351 with the gpd promoter and trpC terminator from *Aspergillus nidulans,* derived from pAN52-1Not [72], was used as a vector for expressing the chimeric and native α-amylase gene variants, while plasmid containing the *A. niger pyrE* marker gene was used as a selection marker for *A. niger* transformation [73]. For culturing the *A. niger* strains the minimal medium (MM) and complete medium (CM) were used [74]. Luria–Bertani medium with ampicillin 100 µg/mL was used to culture *E. coli* DH5α. For the elution of bound protein, the corn-based malto-dextrin from Merck (419672) was used. For enzyme activity assays, several substrates were used: AZCL-amylose (Megazyme), 2-Chloro-4-nitrophenyl α-D-maltotrioside (CNPG3) (Biosynth, EC09787), and various starch granules purchased from Sigma–Aldrich, including soluble starch (S9765), rice starch (S7260), corn starch (S4126), wheat starch (S5127), and raw potato starch (Honig, The Netherlands). All starch substrates used are insoluble. The 3.5-Dinitrosalicylic acid (DNS) solution was also purchased from Sigma–Aldrich (D0550) to conduct enzyme activity analysis based on the DNS assay with starch substrates.

### 3.2. Gene Design for Chimeric CBM20_GH13

In order to construct the new α-amylase with an N-terminal CBM20 (referred to as CBM20_GH13), we made the following amino acid sequence selections based on naturally occurring and nature-inspired configurations: the GH13 α-amylase encoding gene from *A. niger* (database accession ANI_1_460094), the signal sequence and CBM20 encoding sequence from *A. niger* Glucoamylase (CAA25219.1), the linker region joining a CBM20 and GH13 domain from algae *Neopullulanase tetraselmis* (JAC81539.1), and a linker joining the signal sequence and CBM20 from a Fungal Laccase (XP_007679364.1). The signal sequences were predicted using SignalP 6.0 (https://dtu.biolib.com/SignalP-6 (accessed on 20 December 2022)). Protein domain analysis was performed using the HMMER program (Version 3.3.2) based on the conserved domain database available in Pfam [75]. Based on this design, a codon optimised gene version was synthesised at BaseClear, The Netherlands. The molecular weight and theoretical pI were predicted using the ExPASy website (https://web.expasy.org/protparam/ (accessed on 20 December 2022)).

### 3.3. Plasmid Construction, Transformation, Cultivation, and 3D Protein Modelling

For the cloning of the chimeric amylase gene variant, BspLu11I and BamHI restriction sites were added at the 5′terminus and 3′terminus, allowing the chimeric genes to be cloned in pAN52-1Not plasmid [72], where BspLU11l is compatible with the NcoI cloning site of pAN52-1Not.

For generating the expression vector for the GH13 α-amylase without CBM20, assigned as GH13, the vector containing a chimeric CBM20_GH13 gene was used as a template for PCR amplification using primer pairs of AamA_2 and GH13AamARev. Meanwhile, for generating the expression vector of the native α-amylase with a C-terminal CBM20 assigned as GH13_CBM20, the complete gene encoding protein GH13_CBM20 (KAI3001921.1) was amplified from the genomic DNA of *A. niger* AB4.1 using primer pairs of OriAamAf and OriAamA_CBM20r. All final constructs with α-amylase variants were verified by sequencing using primer MBL852 and MBL858 (Macrogen Europe B.V). All the primers details are listed in Table 4. In addition, the prediction of the three-dimensional (3D) models of the GH13 catalytic domain and CBM20 domain were built by Alphafold2 [68,69]. The output from this analysis resulted in several pdb models that were visualised using a protein pdb viewer called PyMOL version 2.5 (http://www.pymol.org (accessed on 20 April 2023)).

The three different expression vectors were amplified in *E. coli* DH5α and each of the vectors used for the transformation of *A. niger* MGG029-Δ*aamA pyrE*^−^. Fungal transformation was carried out according to the protocol described in [74], using the fungal expression vector carrying the gene-encoding α-amylase variants and the *A. niger pyrE* selection marker vector in 10:1 ratio. PyrE^+^ transformants were selected on minimal media containing sucrose as an osmotic stabiliser [73]. Furthermore, 25 randomly selected transformants from each α-amylase expression vector were picked and purified by single colonies streaked on minimal medium agar plates. Subsequently, these 25 purified transformants from each α-amylase construct were streaked onto minimal medium plates containing 0.1% AZCL-Amylose for the screening of α-amylase production. Amylase positive transformants were identified by the formation of a blue halo around the colony. The best performing transformants were selected and used for further cultivation and enzyme characterisation.

For cultivation, submerged fermentation was carried out in 300 mL Erlenmeyer flasks with a 100 mL working volume of liquid complete medium that was inoculated with 1 × 10^8^
*A. niger* spores. These flasks were incubated in a shaker incubator at 180 rpm and 30 °C for 96 h. At the end of the cultivation process, the spent medium containing the secreted enzymatic activities was collected by filtration.

### 3.4. Starch Binding Purification

The starch binding purification was carried out based on corn-starch binding and the elution of protein as described in [49] with some modifications as described below. Prior to binding, 2% (*w*/*v*) corn starch was pre-washed three times using 50 mM acetate buffer (NaAc, pH 5) in a 50 mL tube to remove traces of soluble saccharides. The spent medium was adjusted to pH5 with one third volume 200 mM acetate buffer pH 5. For enzyme binding, the pH 5-adjusted spent medium (45 mL) was mixed in a 50 mL tube which already contained prewashed 2% (*w*/*v*) corn starch and incubated overnight on ice while shaking gently using a rocking shaker (VWR^®^) in a cold room. Eventually, the corn starch with the bound protein was collected by centrifugation (10 min at 3000× *g*, 4 °C) separating it from the supernatant with any unbound proteins. Subsequently, the collected corn starch–enzyme complex was washed once with 50 mM NaAc buffer pH 5 by centrifugation (10 min at 3000× *g*, 4 °C) to remove any unbound protein or nonspecific protein binding. The supernatant and wash solution containing unbound protein were collected for determining the total unbound activity. The percentage of bound protein was determined by assaying enzyme activity in the spent medium and unbound fraction. Binding potential (%) was determined using the following equation:Bound protein (%)=(Initial activity−final activity)/(Initial activity)×100%

Initial activity represents the activity in the spent medium prior to binding, while final activity is the activity of unbound protein contained in both the supernatant and the wash buffer. The bound protein was eluted from the corn starch–protein complex by the addition of 1.25 mL elution buffer (50 mM NaAc, pH 5) containing 2% malto-dextrin [64]. The elution sample was incubated at 40 °C for 1 h while gently mixing at 80 rpm in a shaker incubator (New Brunswick Innova^®^ 44). To obtain the eluted protein, the corn starch was removed by centrifugation (10 min at 3000× *g*, 4 °C), and the supernatant containing purified protein was collected. The malto-dextrin molecules were removed through ultrafiltration with spin columns (Amicon^®^ Ultra-15 Centrifugal Filter Units, MWCO 30 kDa). The malto-dextrin was contained in the flowthrough, and the purified enzyme remained in the retentate. The purified α-amylase in the elution buffer (50 mM NaAc, pH 5) was then stored at −20 °C until being used for further experiments. The evaluation of α-amylase purification was performed by measuring the activity of the spent medium and all the fractions obtained from purification steps using CNPG3 and various starches as substrate (Appendix A).

### 3.5. Enzymatic Activity of α-Amylase

#### 3.5.1. AZCL-Amylose

For the screening of the transformants, the spores of *A. niger* transformants were spotted onto solid minimal medium containing 0.1% AZCL-Amylose (chromogenic substrate) and incubated at 30 °C for 24 h, essentially as described by the manufacturer protocol (www.megazyme.com (accessed on 20 December 2022)) and also as reported in [76]. The amylase positive transformants were identified by the formation of a blue halo around the colony. Furthermore, for the α-amylase activity assays in liquid samples, 50 μL enzyme samples such as spent medium and unbound and eluted protein were mixed with 100 μL of buffer containing 0.1% AZCL-amylose in 50 mM NaAc pH5 in PCR tubes and incubated at 37 °C for 2.5 h. During hydrolysis, the blue colour will be released from the AZCL-Amylose substrate by α-amylase. Subsequently, 100 μL supernatant was transferred to a Microtiter plate, and the optical density at 590 nm was measured. All reactions were carried out in triplicate. To determine the specific activity, 0.5 µg of enzyme was used, and the OD590 was measured at several time points to obtain its specific activity in U/mg,h. The OD590 was plotted in a graph against the time, and specific activity was calculated from the linear part of the curve with the following formula:$$Specific\;Activity=\frac{∆OD\left(Absorbance\right)}{∆time\;(hour)}/mg\;protein=Unit/mg\;protein,h$$

#### 3.5.2. 2-Chloro-4-Nitrophenyl-α-D-Maltotrioside (CNPG3)

The activity of α-amylase on CNPG3 was measured by mixing 50 μL of either spent medium or purification fractions with 50 μL of 10 mM CNPG3 dissolved in 50 mM NaAc pH 5 and incubation at 37 °C for 30 min as essentially described in [77]. After incubation, the reaction was terminated by adding 100 µL of 0.5 M Na_2_CO_3_. Subsequently, the 100 µL of each reaction sample was transferred to a 96-well plate (Type F, Sarstedt), and the absorbance was measured at 405 nm using a plate reader (Tecan Spark^®^ 10 M, Männedorf Switzerland). A standard curve was prepared using 2-chloro-4 nitrophenol (CNP) with a concentration ranging from 0.2 to 1.5 mM. One unit (U) of enzyme activity is defined as the amount of amylase that catalysed the formation of 1 μmol CNP per hour under assay conditions. This activity assay was performed in triplicate.

#### 3.5.3. Dinitrosalicyclic Acid (DNS) Assay

The enzymatic activity of the α-amylase variants was also investigated using several types of starches as substrates, including starch from rice, wheat, and corn, soluble starch, and raw potato starch in a DNS assay [78]. These starches are insoluble granules and were used as suspension. 1% of starch was prepared in 50 mM NaAc buffer pH 5. For the reaction, 50 μL of the enzyme sample was mixed with 50 μL of each of the 1% starch substrates in a PCR tube. After incubation at 37 °C for 1 h, the mixture was centrifuged, and 75 μL of supernatant was transferred into a clean PCR tube and mixed with 75 μL dinitrosalicyclic acid (DNS) solution followed by incubation at 100 °C for 10 min. After incubation, the mixture was treated with 30 μL of 40% potassium sodium tartrate solution to terminate the enzyme reaction. One hundred microliters of mixture were transferred from the PCR tubes into a 96-well plate (Type F, Sarstedt), and the OD540 was measured using a plate reader (Tecan Spark^®^ 10 M). The soluble reducing sugar released was quantified based on a standards curve prepared with 0–10 mM glucose. A blanc without enzyme was included and its OD540 subtracted for all measurements. All experiments were carried out in triplicate. One unit of enzyme was defined as the amount of enzyme that produced 1 μmol of reducing sugar or equivalent to 1 μmol of glucose per hour.

### 3.6. SDS-PAGE and Zymogram Analysis, Protein Concentration

Sodium dodecyl sulfate-polyacrylamide gel electrophoresis (SDS-PAGE) was performed in 10% precast polyacrylamide gels (Bio-Rad, Mini-PROTEAN^®^ TGX™, #4561033, Hercules, CA, USA) as essentially described in [79]. The spent medium sample and the purification fractions were mixed with SDS-PAGE loading buffer (50 mM Tris-HCL pH 6.8, 25% glycerol, 0.05% bromophenol blue, 1% SDS) in a 3:1 ratio, heated for 5 min at 95 °C, and loaded into the precast polyacrylamide gel. As a molecular weight standard, 5 μL unstained marker (Bio-Rad, precision plus protein unstained standard #161-0363) was loaded as well. The electrophoresis was run at 120 V for 1 h at room temperature. Prior to staining, the gel was submerged in 40 mL of fixation buffer (50% (*v*/*v*) MeOH, 7% (*v*/*v*) Acetic Acid) and incubated at room temperature for 60 min while shaking gently using a rocking shaker (VWR^®^) and changed with fresh fixation buffer solution every 30 min. Subsequently, 35 mL of Sypro Ruby staining solution (Bio-Rad, Cat# 1703125) was poured over the gel in a dark container and incubated overnight at room temperature with gentle shaking on a rocking shaker. The gel was destained with 40 mL of destaining buffer (10% (*v*/*v*) MeOH, 5% (*v*/*v*) Acetic Acid). Gel imaging was performed with the BioRad GelDoc^TM^ EZ Imager. Automatic image exposure and band intensity were selected. For the background image the colour gray was selected, while white was chosen for the Sypro Ruby stained protein bands (as shown in Appendix A). The image was exported as a TIFF file with 300 dpi resolution.

For the detection of α-amylase activity through zymography as essentially described in [80], samples were mixed with 1× loading buffer (as described above, without SDS) at 3:1 ratio and loaded into 10% precast polyacrylamide gel. The electrophoresis was run at 80 Volt for 90 min. Subsequently, the gel was overlayed with AZCL-amylose (0.1%) and agarose (0.3%) in 50 mM of NaAc buffer pH5 as described in [81]. This was followed by incubation for 3 h at 37 °C, and the activity was visible directly from the formation of a blue colour at the position of protein bands with α-amylase activity due to the release of the azo dye from the AZCL-amylose. Lastly, the Bradford method was used to determine the protein concentration with the Bradford calorimetric assay (Bio-Rad 5000006) using bovine serum albumin (BSA) as standard (Bio-Rad). The protocol was carried out according to the manufacture instruction manual.

### 3.7. Kinetic Enzyme Analysis

For the kinetic parameters of the purified enzymes as described in [82,83], the Michaelis–Menten constant (Km) and maximum velocity (Vmax) were determined according to the Michaelis–Menten equation by measuring the enzymatic reaction rate per 1 mg of protein toward various substrates at different substrate concentrations until the enzyme reached it saturated point, which ranged from 1–20 mg/mL for CNPG3, 1–10 mg/mL for soluble starch and corn starch, 1–15 mg/mL for rice starch, and 0.5–5 mg/mL for wheat and raw potato starch at 37 °C for 30 min. The data was plotted according to the Lineweaver–Burk method to obtain the Km (mg/mL) and Vmax (µmol/min). The standard curve was made based on the reaction products, 2-choloro-4 nitrophenyl for CNPG3 substrate or glucose for various starch substrates.

## Figures and Tables

**Figure 1 molecules-28-05033-f001:**
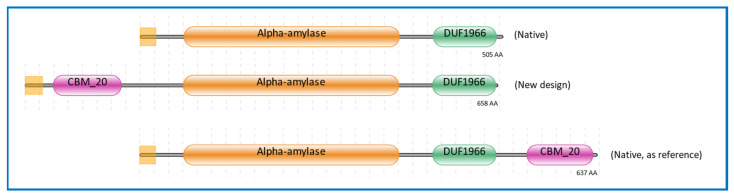
Domain architectures of α-amylases used in this study. The color represents the region of the modules, purple for CBM20 domain, gold for GH13 α-amylase catalytic domain, green for the unknown domain function of DUF1966, and the yellow square at the left for the signal sequence. The source of sequences is listed in Section 3.3.

**Figure 2 molecules-28-05033-f002:**
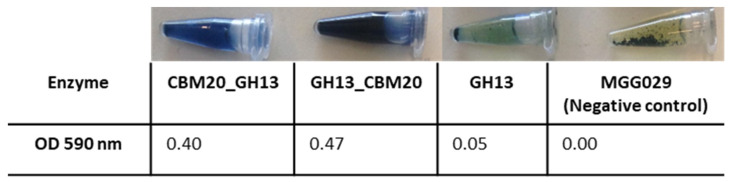
Expression analysis of the α-amylase variants with a CBM. Enzyme activity was measured using AZCL-amylose as substrate and spent medium from liquid cultures of the amylase variants. The negative control is the host strain *A. niger* MGG029-Δ*aamA* transformed with the empty vector.

**Figure 3 molecules-28-05033-f003:**
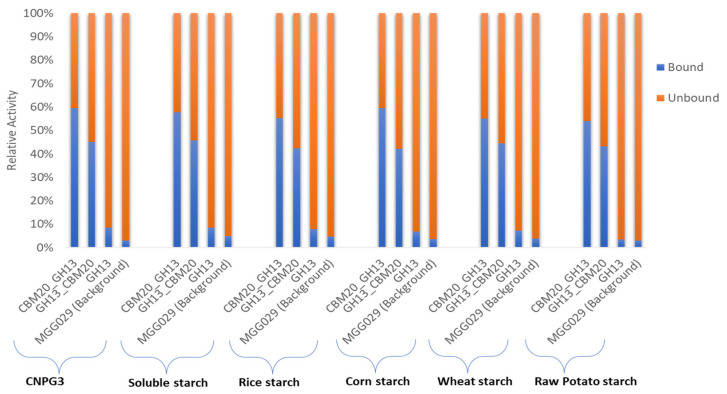
The binding potential of α-amylase variants on corn starch was determined by subtracting the activity in the unbound fraction from the total activity in the spent medium. The activity in the spent medium was set at 100%. The total activities were measured using various substrates and based on nitrophenyl release from the CNPG3 substrate and reducing sugars from the various starch substrates, as listed in Appendix A.

**Figure 4 molecules-28-05033-f004:**
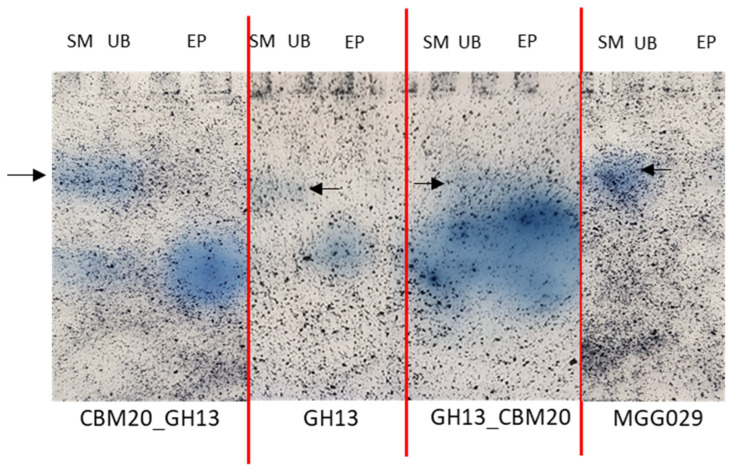
Zymogram analysis of various α-amylase variants on native-PAGE gel stained with AZCL-Amylose substrate. All samples are fractions that were obtained from the starch binding purification: SM is spent medium, UB is unbound and EP is eluted protein. The black arrow indicates a background activity in the SM and UB fractions, also present in the parental host strain *A. niger* MGG029-Δ*aamA*. The same amount of total protein is loaded for each fraction.

**Figure 5 molecules-28-05033-f005:**
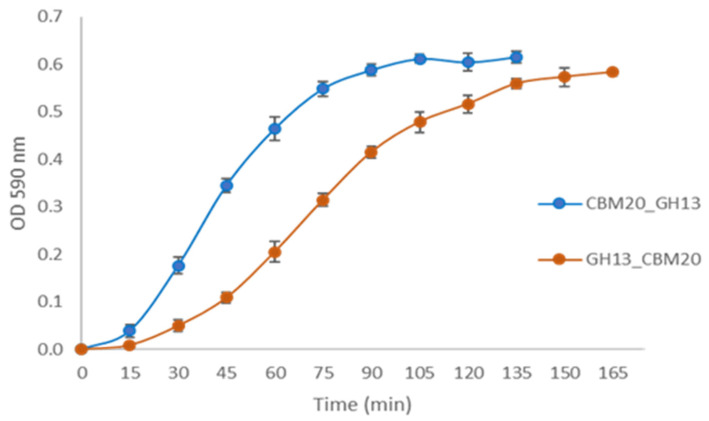
Hydrolytic activity of purified α-amylase using AZCL-amylose as a substrate. The α-amylase was purified through starch binding as described in Materials and Methods Section 3.4.

**Figure 6 molecules-28-05033-f006:**
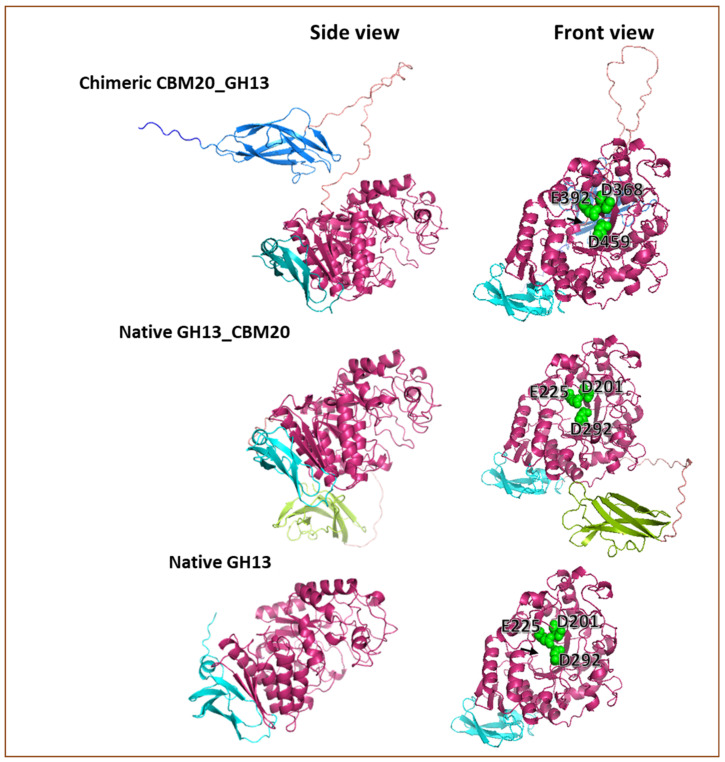
Predicted 3D models of the protein structure of the α-amylase used in this research. This 3D prediction was built using Alphafold 2 [68,69] and visualised with PyMOL [70,71]. The N-terminal and C-terminal CBM20 domain are colour coded dark blue and green, respectively, while the GH13 catalytic domain is pink and the DUF1966 domain is cyan. The black arrow points to the N-terminal end of the GH13 catalytic domain. The catalytic triad in the substrate binding cleft is shown as spheres in green.

**Table 1 molecules-28-05033-t001:** Amino acid sequence of the natural linkers connecting either N or C-terminal CBM20 with the GH13 domain of α-amylase.

α-Amylase	Linker Composition
Chimeric CBM20_GH13	VSQEQWWCSEDDPAAVAASQAARVYMDCHPKPRHPRKPIPVFVPD
Native GH13_CBM20	GSNSSTTTTTTATSSSTATSKSASTSSTSTACTATST

**Table 2 molecules-28-05033-t002:** Specific Activity (U/mg,h) and relative specific activity (%) of purified GH13 α-amylase variants toward various starch substrates. Specific activity on chimeric α-amylase CBM20_GH13 toward each substrate was set as 100% relative activity. The α-amylase was purified through starch binding as described in Materials and Methods Section 3.4.

	α-Amylase	Rice Starch	Soluble Starch	Corn Starch	Wheat Starch	Raw/Native Potato Starch	CNPG3
Specific activity ^a^	CBM20_GH13	1752 ± 18	1126 ± 12	684 ± 3	370 ± 11	177 ± 1	4815 ± 16
	GH13_CBM20	1426 ± 27	902 ± 14	538 ± 6	275 ± 7	84 ± 0.8	3886 ± 33
Relative activity	CBM20_GH13	100 ± 1	100 ± 1	100 ± 0.4	100 ± 3	100 ± 0.6	100 ± 0.3
	GH13_CBM20	81 ± 2	80 ± 2	78 ± 1	74 ± 2.5	47 ± 1	81 ± 0.8

^a^ The experiments were conducted in triplicate. The ± value represents standard error from each experimental point.

**Table 3 molecules-28-05033-t003:** The values of Vmax and Km for CBM20_GH13 and GH13_CBM20 based on various substrates. The α-amylase was purified through starch binding as described in Materials and Methods Section 3.4.

Substrate	Vmax (µmol/min) ^a^		Km (mg/mL) ^a^	
	CBM20_GH13	GH13_CBM20	CBM20_GH13	GH13_CBM20
CNPG3	90.9	71.4	4.2	4.9
Soluble starch	20.3	15.6	3.5	4.5
Rice starch	30.2	22.4	4.0	6.4
Corn Starch	10.3	8.1	2.9	4.6
Wheat starch	6.2	4.2	1.2	2.2
Raw Potato starch	3.5	1.5	0.8	1.6

^a^ The experiment was performed in triplicate. The mean value was used to calculate Vmax and Km. The standard error value from each experimental point was less than ±0.08.

**Table 4 molecules-28-05033-t004:** List of primers used in this study.

List of Primer Name	Sequence 5′ to 3′	Targeted Site
AamA_2 (forward)	TGGCGGACACAATCCATC	GH13 aamA gene of *A. niger* CBS 513.88 from the plasmid containing CBM20_GH13
GH13AamARev (reverse)	GGCCAGACCTGTGCAGAC	GH13 aamA gene of *A. niger* CBS 513.88 including glaA signal sequence from the plasmid containing CBM20_GH13
OriAamAf (forward)	ACATGTCGAGACTATCGACTTCA	Native GH13_CBM20 from *A. niger* AB4.1
OriAamA_CBM20r (reverse)	GGATCCCTACCTCCAAGTATCAACCACC	Native GH13_CBM20 from *A. niger* AB4.1
MBL852 (forward)	GCTACATCCATACTCCA	GPD Promoter until the gene of insert for confirming correct construct
MBL858 (reverse)	ATATCCAGATTCGTCAAGCTG	trpC terminator until the gene of insert for confirming correct construct

## Data Availability

All available data are included in the article.

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
