# Peer review of "Novel Design of an α-Amylase with an N-Terminal CBM20 in Aspergillus niger Improves Binding and Processing of a Broad Range of Starches"

_molecules, 2023, doi:10.3390/molecules28135033_

Round 1

Reviewer 1 Report

Sidar et al report on an original domain shuffling of an a-amylase from Aspergillus niger showing that the unusual architecture with an N-terminal rather than a C-terminal CBM20 domain had improved activity. This illustrates that the naturally rarely occurring architecture with an N-terminal CBM20 for these enzymes is actually having a clear potential for improving the native activity. The work reflects a practical application interest and investigates the activity both on an oligosaccharide substrate CNPG3, soluble starch and various starches, including native (raw) starch. Some points to be considered are listed below.

The manuscript is detailed and generally not reader-friendly and overall not so easy to follow. Moreover, it can benefit from a very careful proof reading. Some relatively few (not exhaustive) examples of places to be corrected are given below.

Major points

1.      You may like to refer to the review on starch binding domains by Janecek et al in Biotechnological Advances, 2019 for example after “..surface” (in line 40).

2.      Please make it clear (line 47) that amylose has only 1-4 linkages (there are though very few branch points in some amyloses, but 1-6 linkages are not characteristic of amylose rather than of amylopectin)

3.      Please make it clear that the enzymes are produced using A. niger as host (if this is correct). Would there be glycosylation of the linker (and the globular domains) in this host? Mostly in fungi linkers are O-glycosylated.

4.      In the text (around lines 132-143) describing the origin of the various domains, it may be uncertain how both the GH13-CBM20 and the GH13 are native. So are they actually two different enzymes having different GH13 domains? It might have been of more systematic value to in fact compare two forms containing the same GH13 domain. In the same place in the text please make it clear if the GH13 of the native enzyme not containing a CBM20 and the designed CBM20-GH13 has the same GH13. The various origins could be informed in the legend to figure 1 to be reader-friendly.

5.       One needs to take the structures in Figure 2 with a grain of salt. Also explain how they are obtained in the Figure legend. It is hard to take this information as evidence for processive action on the substrate (line 174). Note that the active site is not pointed out although in the text it refers to as being exposed. Moreover, as asked above are the linkers glycosylated?

6.      The “spent medium” is lab jargon. It of course makes good sense that the medium from the cultures contains the enzymes as they are designed to be secreted. But you can call that “the culture medium” or “the culture supernatant” or explain in these more conventional terms what you mean by “spent medium”. Is any of the enzymes of interest remaining intracellularly?

7.      In Figure 3 when the activity is tested, this obviously is a crude indication as one does not know what is the actual enzyme concentration in the samples and the specific activities of these enzymes.

8.      In the heading (or footnote) of the table S1, do indicate to which material the culture liquid is added to have the enzyme bound. Is this always corn starch used for the binding? Or? The data in Figure 4 is about the activity towards the different substrates of the bound and unbound fraction of the enzyme are these actually the same as in Table S1? If yes, please say that in the figure legend.

9.      Perhaps it is a bit misleading to say “the other a-amylase variant” when referring to MGG029 (line 264) as MGG029 is not an a-amylase variant but the parent (host?) strain.

10.   This reviewer is not sure that Figure 5 provides information not given elsewhere (Table S1). Would one not expect that the activity was highest in the SM?

11.    Is the lag phase in Figure 6, characteristic of the substrate? Can initial rate of hydrolysis by some other method be obtained for the enzymes to determine the kinetic parameters following the Michaelis-Menten mechanism.  How are the progress curves looking for the other substrates (the starches and the oligosaccharide)? In the kinetics experiments one should know the enzyme concentration preferably in the molar unit system. Please note that Figs. S2 and S3 show the presence of other protein bands in addition to the recombinant target enzyme. This makes a quantification for kinetics analysis problematic. Thus lane 4 in Fig. S2 has probably about 20% of contaminating protein with higher molecular weight than the target protein, and a similar level is seen for lane 2, while lane 3 seems of good purity.

12.   On the substrates (for example in Table 3): are the starches from rice, corn, wheat regular commercial starches and not starch granules, while raw potato starch is actually starch granules? Please make sure to make it clear for the readers which state/type of starch is used as the different substrates.

13.   In principle raw starches and other physical states of starches have similar amylose contents and the lower activity on raw starches can be considered as due to them being in the state of particles (granules).

14.   I wonder if it is true to refer to the CBM20-GH13 as a new domain architecture, as some albeit quite rare cases are found ion nature, but have according to the authors not been characterized.

Minor points

1.      Please include space before the units in many places there is no space. This also includes spaces to be removed at end of sentences certain places. In fact check also carefully for the occurrence of double spaces between words in some cases. Also there is space between A. and niger in A. niger.

2.      Make sure in the list of references that the second word in Latin names starts by lowercase.

3.      Line 11, correct to “industries”

4.      Line 12, correct to “bonds”

5.      Line 15, correct to “a-amylases”

6.      Line 16, correct to “Hydrolase”

7.      Line 37, correct to “...always attached at…”

8.      Greek micron to be corrected for “u”

9.      Line 33, correct to “glycoside”

10.   Please correct to CAZy and CAZymes (not CAZYmes) throughout

11.   Probably “sp.” is correct to include “.” In the abbreviation for species in Latin names of organisms

12.   Line 67, remove “enzymatically”

13.   Line 99, something is wrong here, please rephrase to correct

14.   Line 129, replace “enzymatic” with “catalytic”

15.   Please note that “genes are expressed”, while “proteins are produced”.

16.   When you refer to materials and methods section, do inform the number of the section referred to.

17.   In Tables please try to align the numbers so that the “decimal place” if any or the most right digit align vertically for the different rows.

18.   Line 252 says 42% is bound, is that seen as 44% in table S1?

19.   Heading column 3 table S1 correct to “nitrophenol”

20.   The substrates are not “nitrophenyl of CNPG3” it is the nitrophenylate that is hydrolyzed (line 324)

21.   What is “cracking buffer”? (line 548)

Author Response

Below (in this box) is the response we addressed for the Reviewer 1 comments. For the combined responses to all Reviewers, please see the attached file. 

Response to Reviewer 1 Comments

Major points

Point 1: You may like to refer to the review on starch binding domains by Janecek et al in Biotechnological Advances, 2019 for example after “..surface” (in line 40).

Response 1: Please see response to point 2 of the academic editor.

Point 2: Please make it clear (line 47) that amylose has only 1-4 linkages (there are though very few branch points in some amyloses, but 1-6 linkages are not characteristic of amylose rather than of amylopectin)

Response 2: Manuscript is corrected

Point 3: Please make it clear that the enzymes are produced using A. niger as host (if this is correct). Would there be glycosylation of the linker (and the globular domains) in this host? Mostly in fungi linkers are O-glycosylated.

Response 3: We agree that glycosylation does occur in A. niger and could affect the enzymatic activity. However, we have not taken glycosylation levels into account. In this case, the main purpose is to generate the chimeric fungal alpha amylase CBM20_GH13 and compare it with native fungal alpha amylase including GH13 and GH13_CBM20 as reference. These alpha amylase genes were all expressed in the same host, A. niger MGG029 ΔaamA.

Point 4: In the text (around lines 132-143) describing the origin of the various domains, it may be uncertain how both the GH13-CBM20 and the GH13 are native. So are they actually two different enzymes having different GH13 domains? It might have been of more systematic value to in fact compare two forms containing the same GH13 domain. In the same place in the text please make it clear if the GH13 of the native enzyme not containing a CBM20 and the designed CBM20-GH13 has the same GH13. The various origins could be informed in the legend to figure 1 to be reader-friendly.

Response 4:

In this study we were interested to create a new fungal alpha amylase domain architecture with the CBM20 at the N-terminal of the GH13 alpha amylase. A CBM20_GH13 domain architecture does not exist in fungi nor was it characterized from any CBM20_GH13 producing organism.

As comparison we took the native A. niger alpha amylase GH13 without CBM20 and the native GH13_CBM20 to compare with the engineered design.  In this case, the GH13 domain used for the N-terminal engineered version was kept identical to the native GH13 without CBM to see if the engineered version was active and better than the GH13 only. The native GH13_CBM20 was selected without introducing any amino acid changes avoiding the introduction of more variables in the enzymatic comparison. The interest was to show that the engineered CBM20_GH13 ends up in the same activity ballpark as the GH13_CBM20. The GH13 catalytic domain in native GH13_CBM20 is not identical to the GH13 used in chimeric engineered CBM20_GH13 but their similarity is 99%. For further reference the amino acid alignment is added to the supplementary data, it will be Figure S2 in revised manuscript.

Point 5: One needs to take the structures in Figure 2 with a grain of salt. Also explain how they are obtained in the Figure legend. It is hard to take this information as evidence for processive action on the substrate (line 174). Note that the active site is not pointed out although in the text it refers to as being exposed. Moreover, as asked above are the linkers glycosylated?

Response 5: Yes, we agree that the 3D structures built by Alphafold are only a prediction, by no means proof.  We used the structure predictions to discuss if they align with the experimental results.  Alphafold is added to the Figure 6 legend (in revised manuscript). The paragraph on structure prediction has been moved to the end of the Results and Discussion section to accommodate the concern of the reviewer. Then, the number of Figure will be changed from Figure 2 become Figure 6 in revised manuscript.

Based on the amino acid sequences of the two linkers (Table 1), their  glycosylation  could indeed be different. The linker of GH13_CBM20, consisting of many Ser and Thr residues predicted to be O-glycosylated (Goto, 2007, DOI: 10.1271/bbb.70080), whereas the linked used for the CBM20_GH13 design is not predicted to be. However, as mentioned in response to point 3, we did not study the role of glycosylation into account in this study.

Point 6: The “spent medium” is lab jargon. It of course makes good sense that the medium from the cultures contains the enzymes as they are designed to be secreted. But you can call that “the culture medium” or “the culture supernatant” or explain in these more conventional terms what you mean by “spent medium”. Is any of the enzymes of interest remaining intracellularly?

Response 6: Spent medium is commonly used in many papers. However, to accommodate the concern of the reviewer in material & methods, referral is made to define what spent medium is.

Point 7: In Figure 3 when the activity is tested, this obviously is a crude indication as one does not know what is the actual enzyme concentration in the samples and the specific activities of these enzymes.

Response 7: Yes, Figure 3 was basically meant to show that the α-amylase variants from the construct can be expressed and are active. Note that in revised manuscript, Figure 3 will be changed to be Figure 2.

Point 8: In the heading (or footnote) of the table S1, do indicate to which material the culture liquid is added to have the enzyme bound. Is this always corn starch used for the binding? Or? The data in Figure 4 is about the activity towards the different substrates of the bound and unbound fraction of the enzyme are these actually the same as in Table S1? If yes, please say that in the figure legend.

Response 8: Correction for Table S1: in the footnote is added: *these activities were measured based on the corn-starch binding purificationCorrection for Figure 4 (in revised manuscript will be Figure 3), will be added with “as listed in Table S1”

Point 9: Perhaps it is a bit misleading to say “the other a-amylase variant” when referring to MGG029 (line 264) as MGG029 is not an a-amylase variant but the parent (host?) strain.

Response 9: Manuscript is corrected.

Point 10: This reviewer is not sure that Figure 5 provides information not given elsewhere (Table S1). Would one not expect that the activity was highest in the SM?

Response 10: Figure 5 (in revised manuscript will be Figure 4) provides the relative information related to Table S1. Both Figure 5 and Table S1 are given to evaluate the purification based on starch binding.  As expected, the alpha amylase activity in the spent medium was not the highest because the amylase concentration is lower prior to purification and concentration, therefore in Figure 5 we provide relative data.

Point 11: Is the lag phase in Figure 6, characteristic of the substrate? Can initial rate of hydrolysis by some other method be obtained for the enzymes to determine the kinetic parameters following the Michaelis-Menten mechanism.  How are the progress curves looking for the other substrates (the starches and the oligosaccharide)? In the kinetics experiments one should know the enzyme concentration preferably in the molar unit system. Please note that Figs. S2 and S3 show the presence of other protein bands in addition to the recombinant target enzyme. This makes a quantification for kinetics analysis problematic. Thus lane 4 in Fig. S2 has probably about 20% of contaminating protein with higher molecular weight than the target protein, and a similar level is seen for lane 2, while lane 3 seems of good purity.

Response 11: Figure 6 (in revised manuscript will be Figure 5) suggests that with AZCL-amylose as substrate there is a lag phase. This is not observed for CNPG3 and several other substrates, see Figure S4 (in revised manuscript will be Figure S5).

In relation to the kinetic parameters the reviewer makes a relevant observation. However, for the enzyme concentration in Molar Unit system, it is difficult to provide a correct molar concentration for the purified sample due to the fact that indeed a minor impurity identified as a protein band at 250 kDa is present (Figure S2 and S3. In revised manuscript these two Figures will be changed to be Figure S3 and S4)

We also agree with the reviewer that the higher molecular weight band is roughly at the same intensity in the CBM_GH13 and GH13_CBM samples. In the GH13 sample the 250 kDa protein is more intense because the same amounts of total protein were loaded and much less GH13 was bound during purificationWe considered the CBM20_GH13 and GH13_CBM20 purified samples, being of equal purity to be good enough to make a comparison regarding the purification with corn starch and their enzymatic activity.

Point 12: On the substrates (for example in Table 3): are the starches from rice, corn, wheat regular commercial starches and not starch granules, while raw potato starch is actually starch granules? Please make sure to make it clear for the readers which state/type of starch is used as the different substrates.

Response 12: Correction is made in the manuscript by indicating the source of the starches in Material & Methods.

Point 13: In principle raw starches and other physical states of starches have similar amylose contents and the lower activity on raw starches can be considered as due to them being in the state of particles (granules).

Response 13: Yes, we agree that basically raw starch could be more resistance toward enzymatic degradation than modified starch.

Point 14: I wonder if it is true to refer to the CBM20-GH13 as a new domain architecture, as some albeit quite rare cases are found in nature, but have according to the authors not been characterized.

Response 14: Manuscript is corrected by revised the term “a new domain architecture” with “a new characterized domain architecture”. See also the response on point 4 from academic editor.

Minor points

Response to all minor points: We agree with the suggestion below and these points are corrected in the manuscript.

    1. Please include space before the units in many places there is no space. This also includes spaces to be removed at end of sentences certain places. In fact check also carefully for the occurrence of double spaces between words in some cases. Also there is space between A. and niger in A. niger.

  1. Make sure in the list of references that the second word in Latin names starts by lowercase.
  2. Line 11, correct to “industries”
  3. Line 12, correct to “bonds”
  4. Line 15, correct to “a-amylases”
  5. Line 16, correct to “Hydrolase”
  6. Line 37, correct to “...always attached at…”
  7. Greek micron to be corrected for “u”
  8. Line 33, correct to “glycoside”
  9. Please correct to CAZy and CAZymes (not CAZYmes) throughout
  10. Probably “sp.” is correct to include “.” In the abbreviation for species in Latin names of organisms
  11. Line 67, remove “enzymatically”
  12. Line 99, something is wrong here, please rephrase to correct
  13. Line 129, replace “enzymatic” with “catalytic”
  14. Please note that “genes are expressed”, while “proteins are produced”.
  15. When you refer to materials and methods section, do inform the number of the section referred to.
  16. In Tables please try to align the numbers so that the “decimal place” if any or the most right digit align vertically for the different rows.
  17. Line 252 says 42% is bound, is that seen as 44% in table S1?
  18. Heading column 3 table S1 correct to “nitrophenol”
  19. The substrates are not “nitrophenyl of CNPG3” it is the nitrophenylate that is hydrolyzed (line 324)
  20. What is “cracking buffer”? (line 548)

Reviewer 2 Report

The research presented in this paper explored a novel configuration of GH13 α-amylase modular enzyme by considering various fungal domains such as the catalytic domain, the CBM, and the linkers that connect these different domains. The text appears to be logical and well-organized. The study provides valuable insights into the potential application of the newly designed α-amylase variants in the starch industry. However, critical methodological mistakes and some conceptual arguments should be addressed to validate the results and make the work publishable.

1)           Amylases design:

Why did the authors use different catalytic domains and CBM20?

i) The authors explain that for the native GH13_CBM20, they use the A. niger AB4.1 α-amylase gene without specifying which of the genes from this strain was used (An11g03340 (aamA), An12g06930 (amyA), An05g02100 (amyB), or An04g06930 (amyC). On the other hand, for the construction of the CBM20_GH13, the GH13 catalytic domain was derived from the A. niger CBS 513.88 α-amylase GH13, the same for the enzyme without CBM20.

But the CD sequences shown in Supplementary Figure 1 are the same; consequently, it is necessary to make the appropriate precisions and to include the alignment.

ii) In the case of the CBM20, for generating the native α-amylase with a C-terminal CBM20, GH13_CBM20, the complete gene encoding protein GH13_CBM20 (KAI3001921.1) was amplified from the genomic DNA of A. niger AB4.1. Meanwhile, for the CBM20_GH13, the CBM20 from the A.niger's Glucoamylase (CAA25219.1) was used.

Two different CBMs may produce different hydrolytic and binding characteristics in the resultants' enzymes. Consequently, it is improbable to determine if the observed result is the consequence of the sequence and structure or the CBM position.

2)     3D modeling

The orientation of the CBM20 in the model cannot be considered absolute, i.e., comparison between the enzymes is not entirely realistic, mainly because of the flexibility of the linker. The position of the CBM could be less variable if the substrate is present in the representation. On the other hand, observed results do not depend only on the relative position of the CBM20 to the CD but also on its sequence. So, I wonder what the validity of the comparison is.

3)     Purity of enzymes

Lines 239-241 and 490-493:

Assessment of α-amylase purification cannot be performed by measuring the activity of the eluted protein. The purity of the enzyme must be analyzed by chromatography or electrophoresis. The authors report the performance of SDS-PAGE, but the results shown in Figure S2  are unclear. Could the authors have gotten it wrong and included an agar e DNA gel? My suggestion is to attach the original image of the SDS-PAGE without editing.

4)     Binding efficiency

To make a proper comparison of the "binding efficiency" of enzymes to insoluble starch granules is imperative to have pure enzymes, especially if it is believed that other enzymes that interact with the starch may exist.

Any comparison must start from the same initial conditions for all reactions, that is, the same amount of enzyme in all cases.

An essential point that the authors have to clarify is how to calculate the binding efficiency on soluble substrates such as CNPG3 and soluble starch (Figure 4).

If the authors wanted to report the enzyme's binding to the substrate adequately, they could perform adsorption isotherms with pure enzymes or calorimetry (for example, see the publication by Boraston et al. 2006 JBC 281:587–598).

In line 253, the authors rationalize the fact that not all enzyme was retained by starch binding may be explained by the fact that during binding, the electrostatic interaction of the enzyme toward the starch adsorbent influences the adsorption process as reported for the amylase purification from Bacillus. However, it is possible that it was just necessary to increment the starch concentration or that the insoluble nature of the starch made the particles precipitate and avoid the interaction between the starch surface and the enzymes. The gentle shaking is not enough to maintain the particles in suspension. I insist on the necessity of realizing adsorption isotherms.

5)     Enzymatic activity and kinetic parameter

The kinetic parameters are not related to the protein concentration; even though it is established that the activities are reported per mg of protein, it would be essential to make adequate comparisons to normalize the rates to uM of product/min*uM of enzyme.

In this way, the fundamental kinetic constants Kcat and Kcat/Km can be analyzed.

On the other hand, in Figure 5, the zymogram is meaningless if it is not correlated with protein electrophoresis.

6)     Almost all methods lack references

Other considerations:

In Table S1:

 (A.1.) CBM20_GH13

Volume (ml)

Nitropenol

Line 485:   Please explain the origin and characteristics of the used dextrins

Specify on line 310 the amount of enzyme used in the reactions: The protein concentration of both samples was normalized to...

In the activity, figures report in U.E., not in D.O.

Author Response

Below (in this box) is the response for the Reviewer 2. For the combined responses to all Reviewers, please see the attached file. 

Response to Reviewer 2 Comments

Point 1: Amylases design:

Why did the authors use different catalytic domains and CBM20?

i) The authors explain that for the native GH13_CBM20, they use the A. niger AB4.1 α-amylase gene without specifying which of the genes from this strain was used (An11g03340 (aamA), An12g06930 (amyA), An05g02100 (amyB), or An04g06930 (amyC). On the other hand, for the construction of the CBM20_GH13, the GH13 catalytic domain was derived from the A. niger CBS 513.88 α-amylase GH13, the same for the enzyme without CBM20.

But the CD sequences shown in Supplementary Figure 1 are the same; consequently, it is necessary to make the appropriate precisions and to include the alignment.

ii) In the case of the CBM20, for generating the native α-amylase with a C-terminal CBM20, GH13_CBM20, the complete gene encoding protein GH13_CBM20 (KAI3001921.1) was amplified from the genomic DNA of A. niger AB4.1. Meanwhile, for the CBM20_GH13, the CBM20 from the niger'sGlucoamylase (CAA25219.1) was used.

 Two different CBMs may produce different hydrolytic and binding characteristics in the resultants' enzymes. Consequently, it is improbable to determine if the observed result is the consequence of the sequence and structure or the CBM position.

Response 1 (i) and (ii): Please see response on point 4 from Reviewer 1.

Point 2: 3D modeling

The orientation of the CBM20 in the model cannot be considered absolute, i.e., comparison between the enzymes is not entirely realistic, mainly because of the flexibility of the linker. The position of the CBM could be less variable if the substrate is present in the representation. On the other hand, observed results do not depend only on the relative position of the CBM20 to the CD but also on its sequence. So, I wonder what the validity of the comparison is.

Response 2: These structures are a prediction and built by Alphafold program: We refer to this prediction to support our findings. For the detail response in this 3D modelling topic, please see the response we addressed on point 5 from reviewer 1. 

Point 3:  Purity of enzymes

Lines 239-241 and 490-493:

Assessment of α-amylase purification cannot be performed by measuring the activity of the eluted protein. The purity of the enzyme must be analyzed by chromatography or electrophoresis. The authors report the performance of SDS-PAGE, but the results shown in Figure S2 are unclear. Could the authors have gotten it wrong and included an agar e DNA gel? My suggestion is to attach the original image of the SDS-PAGE without editing.

Response 3: To confirm the purification result, we run the spent medium, unbound fraction, and the eluted protein on SDS-PAGE and zymogram. Both protein gel analyses are electrophoresis-based methods.

Protein the Figure S2 is the original image of an SDS-PAGE. Yes, we understand that the unclear information in the SDS-PAGE visualization methods can result in misinterpretation. Following SYPRO Ruby staining the imaging was done with the BioRad GelDocTM EZ Imager.

For the original pictures, please see the un-cropped gel Figure that was submitted along with the manuscript submission process previously. The staining and brief imaging information will be put to the legends of Figure S2 and S3 (in revised manuscript will be Figure S3 and S4 due to additional Figure in Supplementary as mentioned on point 4 from Reviewer 1).

Point 4: Binding efficiency

To make a proper comparison of the "binding efficiency" of enzymes to insoluble starch granules is imperative to have pure enzymes, especially if it is believed that other enzymes that interact with the starch may exist.

Any comparison must start from the same initial conditions for all reactions, that is, the same amount of enzyme in all cases.

An essential point that the authors have to clarify is how to calculate the binding efficiency on soluble substrates such as CNPG3 and soluble starch (Figure 4).

If the authors wanted to report the enzyme's binding to the substrate adequately, they could perform adsorption isotherms with pure enzymes or calorimetry (for example, see the publication by Boraston et al. 2006 JBC 281:587–598).

In line 253, the authors rationalize the fact that not all enzyme was retained by starch binding may be explained by the fact that during binding, the electrostatic interaction of the enzyme toward the starch adsorbent influences the adsorption process as reported for the amylase purification from Bacillus. However, it is possible that it was just necessary to increment the starch concentration or that the insoluble nature of the starch made the particles precipitate and avoid the interaction between the starch surface and the enzymes. The gentle shaking is not enough to maintain the particles in suspension. I insist on the necessity of realizing adsorption isotherms.

Response 4: We understand that the reviewer addressed the approach for examining binding efficiency. In this research, binding to starch was done to purify the amylase, instead of obtaining the absolute figures for binding itself. Therefore, it was aimed to find the experimental conditions to get sufficient recovery of active enzyme with this binding experiment.  We agree that we better replace “binding efficiency” by “potential binding”.  Overall, this approach was mainly to show that the CBM versions can be purified very well, while GH13 shows less binding potential. We have rephrased the text in the revised manuscript to accommodate the concern of the reviewer

Point 5: Enzymatic activity and kinetic parameter

The kinetic parameters are not related to the protein concentration; even though it is established that the activities are reported per mg of protein, it would be essential to make adequate comparisons to normalize the rates to uM of product/min*uM of enzyme.

In this way, the fundamental kinetic constants Kcat and Kcat/Km can be analyzed.

On the other hand, in Figure 5, the zymogram is meaningless if it is not correlated with protein electrophoresis.

Response 5:  In this case, no good enzyme molar concentration calculation is possible due to high molecular weight impurity (See also response on Point 11 from reviewer 1). Basis of a Zymogram is electrophoresis. In the Zymogram, the same amount of total protein is loaded for each purified sample.  This information will be put in the legend of Zymogram figure.

Point 6: Almost all methods lack references

Other considerations:

In Table S1:

 (A1.) CBM20_GH13

Volume (ml)

Nitropenol

Line 485:   Please explain the origin and characteristics of the used dextrins

Specify on line 310 the amount of enzyme used in the reactions: The protein concentration of both samples was normalized to... In the activity, figures report in U.E., not in D.O.

Response 6: Correction is made in the manuscript.

Reviewer 3 Report

This article generated and investigated the unique domain architecture of an A. niger α-amylase carrying an N-terminal CBM20. The effect of this new domain architecture on the substrate binding properties, catalytic activity as well as kinetic parameters were reported. There are some questions listed below.

1. Why do expression element optimization first instead of kinetic analysis to determine the optimal fusion protein?

2. What causes the difference in the properties of the two fusion proteins? The analysis of this mechanism is not clear enough in this article.

3. The analysis of the properties of the two fusion proteins acting on starch substrates is not enough to show the advantage.

4. How to evaluate the accuracy of 3D protein modelling results? Whether the results of different software modelling have been compared?

5. The format of some units is not standardized and unified, such as "ml", "ug", "hours", "h", and there is a lack of space between some units and numbers. Please pay attention to correct it.

Author Response

Below (in this box) is the response we addressed for the Reviewer 3 comments. For the combined responses to all Reviewers comments, please see the attached file. 

Response to Reviewer 3 Comments

Point 1:  Why do expression element optimization first instead of kinetic analysis to determine the optimal fusion protein?

Response 1: We did not conduct the expression optimization. The gene expression analysis was carried out to determine whether the introduced alpha amylase gene is expressed successfully as an active enzyme by A. niger.

Point 2: What causes the difference in the properties of the two fusion proteins? The analysis of this mechanism is not clear enough in this article.

Response 2: In this research we only created one fusion protein, the chimeric CBM20_GH13, instead of two fusions. The GH13_CBM20 used in this research is a native alpha amylase that originally exist in A. niger (Please see response on point 4 from Reviewer 1).

Point 3: The analysis of the properties of the two fusion proteins acting on starch substrates is not enough to show the advantage.

Response 3: See previous point. The aim of our research is to show that the engineered domain architecture of CBM20_GH13 is active and works better than GH13 alone. Furthermore, we wanted to show that the CBM20_GH13 is in the same activity range as the reference alpha amylase GH13_CBM0, to allows us to conclude that an N-terminal CBM is a good option when designing functionally relevant alpha amylases.

Point 4: How to evaluate the accuracy of 3D protein modelling results? Whether the results of different software modelling have been compared?

Response 4: Please see response on point 5 from reviewer 1.

Point 5: The format of some units is not standardized and unified, such as "ml", "ug", "hours", "h", and there is a lack of space between some units and numbers. Please pay attention to correct it.

Response 5: Manuscript is Corrected.

Round 2

Reviewer 1 Report

Sidar et al have made considerable improvements to their manuscript on an unusual domain architecture engineered for an a-amylase from Aspergillus niger having an N-terminal rather than a C-terminal CBM20 domain. This new architecture was accompanied by improved activity. It is of course a major drawback that the characterization is not done for pure proteins and that situation has to be very clearly described to readers and also justified directly in the written manuscript.

Examples of some remaining concerns are given below.

Major points

1.       Please make to discuss for readers which impact O- and N-glycosylation, which is very likely to occur when using A. niger as host, may have on activity and stability.

2.       In Table 1 please list the original species / strains of the different domains. This is also important for figure S2a,b. In the legend to figure S2, to use the term “partially conserved” is not appropriate. This goes for the three sequences aligned only, and they are an extremely limited sample of these enzymes. In fact essentially only the three catalytic site residues are invariant in the a-amylases. Explain better therefore what is meant by partially conserved and what is the importance and relevance of this focus.

3.       Please describe if the different starches are granular or not, do they go in solution when used as substrates? The discussion of the granule size of rice starch may be obsolete if it is not native rice starch that is used as substrate. How about the other starches (except for potato starch).

4.       In the abstract correct to inform readers that there are in fact N-terminal CBM20 found with some a-amylases.

5.       Line 40, “polysaccharide surface” is not a proper term, unless the polysaccharide is actually in particle state.

6.       It’s true that Asp, Glu and Asp are invariant, but in fact they are catalytic site residues and mentioned in that capacity, they are in other words more that “just” active site residues.

7.        “liquid AZCl-amylose”, is not the correct term, why liquid is this in solution, is it a suspension, or?  (line 200). Moreover, use the correct AZCL (not AZCl) abbreviation throughout.

8.       The activity unit “U/mg,h” cannot be understood.

9.       What is meant by analyzed for “1 mg” (line 345). Is it referring to unit per mg, or is actually 1 mg used?

10.   For figures S3 and S4, do inform in the legend by which method the a-amylase bands are identified and add molar mass values for the markers in figure S4.

Minor points

1.       Line 16, correct to “Hydrolase”

2.       Note some reference numbers are given as superscript and some as normal font

3.       Make sure to have space between the digit and the unit (e.g. a space before “kDa”, line 288 and elsewhere). Numerous places this is not the case. Also A. niger needs to have a space after A.

4.       Line 73, correct to catalytic

5.       Should “Algae” or “algae” be used? Please be consistent

6.       The signal peptide is not quite termed a domain (line 120)

7.       Line 186, correct to “span”

8.       Consistently use “L” in volume abbreviations, certain places “l” is used.

9.       Please for elution from starch refer to “malto-dextrin” as also the name used in materials, dextrin is not sufficient here.

10.   Figure 3, there is a problem with the legend placement and number

11.   SDS-PAGE has to be consistently hyphenated

12.   What is “GH13 minus CBM20” ? (line 289)

13.   Make sure in the list of references Latin names are in italics if this is the style preferred by the journal.

14.   In Table S1 (and maybe elsewhere) define what is “MGG029”

Reviewer 3 Report

The authors have addressed all my comments in this revision.

Author Response

Thank you for your comment.